# The Association between Physical Activity and Selected Parameters of Psychological Status and Dementia in Older Women

**DOI:** 10.3390/ijerph18147549

**Published:** 2021-07-15

**Authors:** Angelika Cisek-Woźniak, Kinga Mruczyk, Rafał W. Wójciak

**Affiliations:** 1Department of Dietetics, Faculty of Physical Culture in Gorzow Wielkopolski, Poznan University of Physical Education, 66-400 Gorzow Wielkopolski, Poland; k.mruczyk@awf-gorzow.edu.pl; 2Department of Clinical Psychology, Poznan University of Medical Sciences, 60-535 Poznan, Poland; rafwoj@ump.edu.pl

**Keywords:** physical activity, cognitive function, older women, neurodegenerative diseases, 5000 steps

## Abstract

Physical activity has an unquestionable impact on broadly understood human health. One interesting issue related to this is the importance of movement on mental health and cognitive functioning. Research shows that regular physical activity improves the cognitive functioning of adults and people with mental disorders. Regular physical activity can be an important and powerful protective factor in cognitive impairment and dementia in the elderly, and exercise is an important non-pharmacological treatment for mild cognitive impairment or neurodegenerative diseases. This study aims to present the impact of physical activity on selected cognitive functions in physically active women over 60 years of age. The research was carried out in a group of 110 generally healthy women from the area of western Poland over 60 years of age, who were divided into four groups based on the intensity of their physical activity. A pedometer (sport watch) and a physical activity diary were used to measure physical activity. Body Mass Index was assessed. Selected cognitive functions were assessed using the MMSE test, motor and psychomotor skills were measured, and Luria’s auditory memory test and recall test, a clock drawing test, and a GDS test were performed. There were statistically significant relationships between the level of physical activity and the effectiveness of cognitive processes. These results show that about 5000 steps a day is enough to see a positive effect on the mental health and cognitive functioning of this group of the elderly population. The women had an average BMI of 28.1 ± 4.7. BMI, indicating an overweight condition (over 30 kg/m^2^), was observed in 31% of women. The results of this study lead the authors to conclude that physical activity positively influences cognitive function and can be recommended for all seniors who do not have other serious comorbidities that would prevent them from playing sports.

## 1. Introduction

Physical activity is essential in the prevention of many diseases, including chronic diseases. Meta-analyses of several dozen epidemiological studies show that regular physical exercise reduces the mortality rate caused by cardiovascular diseases by approx. 30–50% and the risk of general mortality by approx. 30%. Regular exercise has also been shown to prevent high blood pressure, diabetes, obesity, osteoporosis, cerebrovascular disease, and ischemic heart disease, as well as some types of cancer. Physical movements also have a positive effect on the body of the elderly, favoring successful aging [1].

For many years, more and more attention has been paid to the aspect of the influence of physical activity on the maintenance of well-being and mental health as well as proper cognitive functioning [2]. Properly selected physical activity has a positive effect on quality of life and ensures well-being, while also enabling the elderly to have better control and independence thanks to greater fitness and physical endurance [3]. It also improves their emotional state, including a reduction in anxiety states and improvement in health during depression [4]. This positive effect occurs due to the fact that systematic exercise regulates the stimulation of the sympathetic nervous system and the reactivity of the hypothalamic–pituitary–adrenal axis. Besides this, it promotes neurogenesis and ensure the correct level of neurotransmitters in the brain, including serotonin, dopamine, endorphins, and norepinephrine [5].

The key institution in defining both individual and collective goals related to physical activity and diet is the World Health Organization, which addresses their developed guidelines to either the general population or specific age groups. In terms of physical activity, the WHO recommendations of 2010 indicate it as a tool of primary prevention. For adults over 65 years of age, physical activity should include recreational, leisure, travel, and physical activity (e.g., walking or cycling); work (if the person works); household activities (household chores); fun, games, and sports; or planned exercise in the context of daily family and social activities. To improve cardiorespiratory fitness, muscle strength, bone health, and functional health, and to reduce the risk of NCDs, depression, and cognitive decline, it is recommended that you exercise at least 150 min of moderate-intensity aerobic exercise or at least 75 min of vigorous aerobic exercise per week, or an equivalent combination of moderate and vigorous intensity activities. It should also be said that in virtually all age groups, the benefits outweigh any harmful consequences of implementing regular physical activity and an active lifestyle—at the recommended level of moderate-intensity physical activity of 150 min per week. In the population approach, to reduce the risk of motor organ injuries, society should be encouraged to gradually increase the intensity of physical activity—from less to more intense forms of exercise [6]. If the elderly cannot, due to their health condition, perform the recommended physical activity, they should be as physically active as their skills and conditions allow. This is a particularly common and important situation in the group of adults 65+ [7].

As indicated above, physical activity has a beneficial effect on the mental health of a person, which is, according to the definition of the WHO on well-being, a state in which a person realizes his abilities and can cope with various life situations, can participate in social life, and work productively [8]. Important components of mental health include possessing basic social and cognitive skills; the ability to recognize, show, and shape their own emotions, including having compassion towards other people; flexibility and an ability to deal with unfavorable life situations; the ability to perform functions in social roles; and a harmonious relationship between the body and mind. These factors influence the state of internal psychological harmony to a different degree [9]. Research shows that deliberate and conscious physical activity has an impact on mental well-being because it is, by definition, health-oriented and engenders a high quality of life [10].

An important disease in modern civilization is depression, which is generally defined as a state of temperament characterized by a feeling of depression, sadness, a sense of irresponsibility, decreased activity, and other similar symptoms. Each mood disorder in which the above symptoms become extreme and intensify is considered to be depression [11]. Epidemiological studies show that people who take part in physical exercise or remain active are less prone to developing depression [12]. In men aged 23–27, Paffenbarger et al. observed a reduced risk of depression of 28% in those with high physical activity (>2500 kcal per week) and by 17% in people exercising moderate physical activity (1000–2000 burned calories per week) compared to people with low physical activity [13]. During 8 years of research, Farmer et al. found that the risk of clinical depression was twice as high in women with low physical activity [14]. Biddle and Mutrie summarized in a review article 10 key randomized controlled trials [15]. Based on a meta-analysis of available studies by Craft and Landers [16], who used exercise as a therapy for clinical depression, and the publication of Biddle and Mutrie, it can be concluded that physical activity is associated with a lower risk of developing depression. Experimental studies show that both aerobic and resistance exercises are effective in treating depression, and the training effect is comparable to psychotherapeutic interventions.

Physical activity along with leading a generally active lifestyle in the form of participation in cultural, dance, and craft workshops increased self-esteem, supported psychosocial development and helped restore emotional balance [17]. Sertel et al. conducted a study on a group of 3041 people divided into three age groups: adults aged 18–45 (adults), middle-aged people aged 45–64 (middle-aged), and elderly people aged over 65 (elderly). Quality of life improved in all age groups, and the highest percentage of physically active people was reported in adults. It was concluded that because physical activity alleviates anxiety and depression symptoms, it should be promoted in people of all ages as a healthy habit and lifestyle [18].

I-Min Lee et al., in their prospective cohort studies of over 16,000 healthy, 72-year-old women from the United States, showed that only about 4400 steps/day was significantly associated with lower mortality rates compared to about 2700 steps/day. With more steps per day, the death rate gradually decreased before reaching a level of approximately 7500 steps/day. Stride intensity was not clearly associated with the lower mortality after adjusting for the total number of steps per day. These findings may serve as an incentive for many sedentary people who find 10,000 steps a day an unattainable goal [19].

Nibbeling et al. found that anxiety states are more severe in overweight or obese women and in men with a third-degree of obesity compared to people with a normal Body Mass Index (BMI) [20].

In their studies, Vancampfort et al. noticed that people who do not perform the recommended minimum physical activity per week (150 min) report a lower quality of life in terms of physical, psychological, social, and environmental functioning [21].

Cognitive processes are processes that take place within the nervous system and consist of receiving, processing, and storing information and bringing it back to the external environment according to the pattern reaction → behavior. The basic cognitive processes are attention, memory, perception, and cognitive control/executive functions, and complex processes include thinking and language. Attention is responsible for the selection of information and is closely related to perception, which is necessary for the process of receiving information from the environment and is related to mental modalities. Memory is understood as the ability to extract and store information, while executive functions are defined as the mental activity directing the course of cognitive processes [22].

Research shows that both regular physical activity and intense single training sessions allow for better results in a variety of cognitive tasks. The increase in cognitive abilities as a result of physical activity depends on age—a moderate to severe effect has been proven in adults (young and old), and less in children. Moreover, research suggests that physical activity positively influences verbal functions, which facilitates the learning of words in a new language, leading to the formation of richer networks of words and their new meanings, as well as improving the understanding and spelling of the mother tongue language and detecting syntax errors [23].

Although studies have shown the protective role of physical training in reducing the risk of cognitive impairment and dementia in people of all ages, its prophylactic effect is extremely important in the elderly and this group is the focus of scientists’ attention [24].

Kramer et al. conducted research on the influence of physical activity on cognitive functions in the elderly. The respondents were divided into three age groups without taking into account their physical and mental health: young-old (55–65 years old), middle-old (66–70 years old), and old-old (71 years and older). Elderly people who did mixed resistance and aerobic training had greater improvements in cognitive function compared to those who only exercised aerobically. Moreover, participation in training of shorter duration had the same benefits as participation in training of medium duration. However, it was the long-lasting programs that turned out to have the greatest positive effect on mental performance. Interestingly, the people classified as middle-old benefited the most from the training. The tests of cognitive functions were divided into four groups: executive, cognitive control, spatial, and speed. In each of the tests, the group performing physical exercises achieved much better results than the inactive group [25].

Gates et al. created a research group of people aged 69–95 with mild cognitive impairment. They were divided into groups that performed aerobic and resistance exercises and both types of activity simultaneously. One training session lasted 30–90 min, and training sessions took place 2–4 times a week over 6–52 weeks and varied in intensity (moderate to high). Performing aerobic exercise was closely related to an increase in verbal fluency but not to other executive functions and memory improvement. Memory improved only in the people who performed strength training [26].

Further studies on the influence of physical activity on the process of mild cognitive impairment and dementia were carried out by Heyn et al. Participants aged 66–91 were asked to participate in 30 different tests involving various types of physical activity, such as walking, isotonic exercise, aerobic dance classes, and riding a stationary bike. A single training session lasted 20–150 min (45 min on average), with a frequency of 1–6 times a week, while the duration of the entire study was 2–112 weeks (on average 23 weeks). Meta-analyses of the obtained results confirm the positive effect of regular physical exercise on improving health in people suffering from dementia [27].

Laurin et al. found that physical activity is an essential tool to counter cognitive decline. This relationship was mainly observed in women and was shown to reduce the risk of dementia with increasing levels of physical activity [28]. It was also proven that physical exercises, especially those of an aerobic nature, affect cognitive functioning in people diagnosed with schizophrenia, mainly in the area of working memory and attention, and social cognition [29].

Old age is a natural stage of life in which there are significant changes in the physical and mental spheres as well as human functioning in society. Due to progressing medicine, the invention of drugs for many troubling ailments, people’s lives have significantly extended. The World Health Organization (WHO) thinks that by 2050 people over the age of 60 will make up 22% of the world’s total population—twice as much as today. The aging of the body is a long process, which acts against the will of man. It affects, among other things, the decline in the capacity for intense physical activity, as well as the degeneration of cognitive functioning. However, by maintaining physical activity, these naturally progressing processes can be delayed.

The authors of this study aimed to assess the influence of physical activity on selected cognitive parameters of women over 60 years of age.

## 2. Materials and Methods

### 2.1. Study Participants

The survey was conducted among senior women who declared that they take part in above-average physical activity. The study group consisted of 110 women aged 60–91, in good health (no diagnosed neurodegenerative, no medical contraindications), recruited at information meetings among participants of the classes of the University of the Third Age (UTA) in Gorzow Wielkopolski (western Poland), including voluntary sports activities. In total, 95% of the UTA participants were women; therefore, the study was conducted on a group of women. Moreover, when selecting the female sex as the research group, the differences in aging between men and women, as well as differences in their lifestyle, metabolism, and regenerative processes were taken into account.

The following qualification criteria were adopted: gender—women; age—over 60 years of age; functional status—independence in carrying out everyday activities and good general health. One hundred and fifty women (aged 60–91) responded to the information and participated in the first screening interview, which looked at questions about the prevalence of neurodegenerative diseases, medical contraindications, and physical activity. Ultimately, one hundred and ten women were selected according to the above eligibility criteria. All the ladies were in good health, did not suffer from burdensome chronic diseases, and did not show diagnosed cognitive disorders or other abnormalities that could affect the course of the study.

At the beginning of the research, the women were interviewed regarding their socio-economic status and anthropometric measurements were taken of their body weight and height. Body weight was measured with the use of a medical scale with an accuracy of 1 g, and height with the use of a medical altimeter with an accuracy of 0.1 cm. Based on the weight and height, the body mass index BMI was calculated. According to the guidelines of Babiarczyk and Turbiarz [30], a BMI value of 24–29 kg/m^2^ was adopted as the normative range for the elderly. Among the respondents, 37% were people with higher education, and 37% had a history of smoking (16% of them are active smokers). The women had an average BMI of 28.1 ± 4.7. A BMI indicating being overweight (over 30 kg/m^2^) was observed in 31% of the women.

The study procedure was reviewed and approved by the Bioethics Committee of the Medical University in Poznan, decision number 184/18 of 1 February 2018. All study participants gave their written informed consent. All procedures were carried out according to the ethical standards of the Helsinki Declaration of 1975.

### 2.2. Assessment of Physical Activity

The assessment of physical activity was based on two independent, simultaneously applied methods [31].

The first method measured the number of steps with a GARMIN Forerunner 35 HR sports watch (pedometer), equipped with a step counter. Each person was required to wear the watch on the wrist of their left hand for 7 days. The examined person wore the watch all the time, excluding bedtime and before contact with water (bathing, swimming, and exercising in water). The second method was the assessment of physical activity using an activity diary, in which the participants of the study recorded for the same 7 days what type of activity they did and how many minutes they spent on this activity. Organized sports activities, such as gymnastics, fitness, swimming, dancing, tennis, and Nordic walking, were taken into account, but also other physical activities, such as gardening, cycling, and intense walking. The test was repeated three times at two-week intervals for each participant.

Physical activity was classified based on the number of steps taken during the 7 days, measured with a sports watch and the number of minutes indicated in the physical activity diary, which participants also completed for 7 days. The following were adopted as measurable units: the number of steps per day (data obtained from the pedometer) and the number of minutes per day spent on physical activity by the participants of the study (data from the diary). Participants were also asked not to change the activity undertaken by recording and wearing the device during the registration days. The intensity of physical activity was expressed as the quotient of steps and minutes measured during one day. It is an observational study with no intervention.

### 2.3. Assessment of Psychological Parameters and Dementia

The assessment of the mental state and motor functions of the examined women was carried out with the use of Polish adaptations of psychological tests.

The assessment of the emotional state and the possible occurrence of depression was performed using the Geriatric Depression Scale (GDS). This scale consists of 30 items. A score above 10 points indicates slight depression, and above 20 points, deep depression [32].

The assessment of the presence of cognitive abnormalities, and thus the possible onset of dementia symptoms, was performed using the Mini Mental State Examination (MMSE) [33]. This scale consists of various tasks assessing basic cognitive functions, for which appropriate points are assigned from 0 to 30. For cognitive disorders, without dementia, the range is 24–26 points, and for mild to profound dementia, points below this range. The correct result is a range above 27 points.

The MMSE test is complemented by the classic clock drawing test. This study used a shortened version of the refilling of 3 dials described by Tokko et al. [34]. Correct filling of the targets allows the participant to obtain 3 points, while the lack of correctness, 0 points.

The test of 10 words of auditory memory according to Łuria [35] was also used. The total sum of remembered and repeated words in 5 test samples was given as the result. Moreover, the possibility of restoring the remembered words was assessed after a lapse of 30 min in which the subject’s attention was distracted by performing other tasks. The result is the sum of the repeated words in the recall test.

The assessment of motor and psychomotor functions was determined using the assumptions for the respective subtests from the dementia scale [36]. For assessment of motor speed, the examined women were asked to touch the first two fingers of the non-dominant hand as widely and as quickly as possible. The result is the number of clicks in 5 s.

Psychomotor speed—subjects were asked to make the following movements with their non-dominant hand as soon as possible: (1) clench the hand into a fist on a flat surface; (2) place your palm flat on the surface with the palm down; and (3) place your hand perpendicular to the flat surface with your little finger on your side. After the sequence was demonstrated, subjects were allowed to perform the exercises twice. The result was the number of correctly repeated complete sequences within 10 s.

### 2.4. Statistical Analysis

The normality of the data distribution was tested using the Shapiro–Wilk test. The distributions of the physical activity results, both the number of steps and the minutes per day, were normal. Pearson’s correlation analysis showed a highly significant positive correlation between these parameters (r = 0.83). It was decided to use the objective method, with the indirect number of steps, as the basis for the division of the groups of the studied women in terms of physical activity. For this purpose, a cluster analysis was carried out, which distinguished two groups with significantly different physical activity (<2000 steps and >2000 steps). The analysis of clusters in the second, larger group distinguished three additional groups with significant differences in physical activity (2080–3120 steps, 3200–7450 steps, and 8483–16,750 steps). As a result, the examined women were assigned to 4 groups (G1—the lowest activity, and G2, G3, and G4—the highest activity) that were statistically different in terms of physical activity.

The remaining analyses depended on the normality of the distribution. Comparisons of the means for data with a normal distribution were made using Student’s *t*-test, while in the case of the distribution deviating from the normal, the Wilcoxon test. Correlation analysis of data not subject to a normal distribution was performed using the Spearman’s rho test. The chi^2^ test was used to compare the data distributions against the normative/reference values.

The studied data were presented using basic descriptive statistics, such as the arithmetic mean, standard deviation, range, and median.

All analyses were performed using the statistical program Statistica ver. 13.3 software package (StatSoft Inc., Tulsa, OK, USA), using a significance level of 0.05.

## 3. Results

The results of the study are presented in Table 1, Table 2, Table 3 and Table 4.

The women included in this analysis had a mean (SD) age of 67.3 (6.1) years (range, 60–91 years), who were divided into four groups (G1—the lowest activity, and G2, G3, and G4—the highest activity) that were statistically different in terms of physical activity. The physical activity assessed based on daily steps and minutes spent showed that the women took an average of 4462 steps during the day and spent an average of 92.8 min during the day on physical activity. Regarding the group division in terms of physical activity, it was shown that the women in the group with the lowest activity performed on average 1245 ± 336 steps a day and spent 35.6 ± 10.7 min a day on physical activity.

The group of women with the highest activity (G4) performed an average of 11,335 ± 2434 steps a day and devoted 178.9 ± 35.9 min a day to physical activity. The middle groups (G2 and G3) performed 2602 ± 336 steps and 68.5 ± 17.4 min per day for activity, and 4899 ± 1197 steps and 109.4 ± 33.2 min, respectively.

The physical activity intensity index (steps per minute) among the research group had an average value of 45.3 ± 13.8, while in individual groups of physical activity, the value of the index in the group with the lowest activity was 36.1 ± 8.0, while in the group with the highest activity it was 59.3 ± 19.2 (Table 1).

In terms of the number of women in the groups, the largest group was assigned to the two middle groups with the highest intensity of physical activity (G2–G3), 32 (29.1%) and 38 (34.6%) women, respectively. The smallest group of women with the highest intensity (G4) was 17 women (15.4%). Participants with the lowest intensity of physical activity constituted a group of 23 women (20.9%). In terms of the BMI value in the entire group of participants, the examined mean was 28.1 ± 4.7. The highest value of BMI was recorded in the women in the group with the lowest physical activity (G1) 30.3 ± 4.9, and the lowest in the group with the highest activity (G4) 26.3 ± 2.9 (Table 1).

The statistical analysis showed several significant differences in individual groups of physical activity. The differences concerned the value of the BMI between women with the lowest and the highest physical activity (G1–G3, G1–G4, and G2–G4). Similarly, significant statistical differences were observed in the steps and minutes values as well as the physical intensity index (G1–G3, G1–G4, G2–G3, G2–G4, and G3–G4). In individual groups of physical activity, the women did not differ significantly in age, but the lowest mean age of active women was in the group with the highest activity (G4) 65.6 ± 3.6 (Table 1).

Table 2 shows the results of the selected tests and the significant differences observed in terms of the assessment of cognitive functions, dementia, and mental state of the surveyed women in particular groups of physical activity.

The results of all the tests performed differed significantly in the individual groups of women involved in specific physical activity. The differences between the group with the lowest activity and the groups with the highest activity (G1–G3 and G1–G4) were particularly noticeable. The average result of the GDS test for the entire research group was 3.9 ± 3.3 and, although it did not show the existence of depression, it is interesting that the lowest average result was shown by the group of women with the highest activity (G4) 2.5 ± 2.5, and the highest by the group with the lowest activity (G1) 4.3 ± 3.3.

The mean result of the cognitive impairment test (MMSE) for the entire research group of women was 27.3 ± 2.4. The highest result was recorded in the group of women with the highest activity (G4), 28.5 ± 1.6, and the lowest in the group with the lowest physical activity (G1), 26.1 ± 2.5. In the motor speed and psychomotor speed tests, the mean results obtained were 20.6 ± 4.3 and 5.7 ± 1.8, respectively. The highest results in the above tests were obtained by the women from the G4 group, 22.3 ± 4.1 and 7.0 ± 1.6, while the lowest results were obtained by the women from the G1 activity group, 19.7 ± 5.0 and 5.0 ± 1.9.

The test assessing the effectiveness of direct and delayed auditory memory showed that women in the entire research group learned 36.1 ± 6.2 words in the first five trials and 5.8 ± 2.4 words in the delayed memory test. The women from groups G2, G3, and G4 learned the most words in each activity group, 36.5 ± 5.1, 37.5 ± 6.8, and 37.1 ± 4.6, respectively. The women from the group with the lowest G1 activity, 32.5 ± 6.1, learned the fewest words.

The results of the deferred memory test in individual groups of physical activity showed that the women from groups G3 and G4 had the highest scores, 6.9 ± 2.2 and 6.8 ± 1.7, and the lowest number of women to learn a significant number of words were from the group with the lowest activity (G1), 4.3 ± 2.0.

The mean result of the clock drawing test, i.e., the test complementing the MMSE, was 2.7 ± 0.7. However, the lowest result was achieved by the participants from the group with the lowest activity (G1) 2.3 ± 0.9, and the highest result from the group with the highest physical activity (G4) 2.8 ± 0.7.

Table 3 shows the interrelationships between successive independent variables and the results of the depression and dementia tests performed among the studied women. This analysis shows that over 90% of the women from all groups of physical activity did not have depression (GDS test), and the remaining number of them showed symptoms of “mild depression”, while in the highest activity group, 100% of the women obtained the result “without depression”. The MMSE test showed that in the groups with higher physical activity (G2, G3, and G4) no symptoms of dementia were found in 66%, 61%, and 88% of the women, respectively. In the G4 group, only 12% of participants had cognitive impairment without dementia. The lowest percentage of women without dementia was found in the group with the lowest physical activity (G1), which also had the highest number of mild dementias—22%. Similar values were shown by the clock drawing test, in which the groups of women with the highest physical activity (G3 and G4) obtained correct results in 95% and 94%, respectively, while the most errors were found in women in the groups with less activity (G1 and G2).

Table 4 shows the correlation between physical activity expressed in the number of daily steps performed and the intensity of activity, and the BMI and other assessed parameters of the mental health of seniors. The data showed a fairly strong correlation between physical activity and the onset of depression (GDS) and a weak correlation between auditory memory (Luria’s auditory memory test) and physical activity. In the other assessed parameters, no correlation between physical activity and its intensity was found.

## 4. Discussion

The women in the groups with the highest physical activity (G3 and G4) obtained generally better results in the tests performed and the parameters measured.

The study by Lee et al. [19], cited above, showed that physical activity reduces mortality among the elderly if it is maintained at an appropriate level. This study showed that activity of 4400 steps per day significantly reduced the mortality rate, and activity below this level did not guarantee lower mortality in the study group. Thus, the authors showed that in order to live longer, a goal of 10,000 steps is not necessary, despite the fact that this is often conveyed to the public with a limited scientific basis.

The results of this study showed that there is a clear limit to the amount of physical activity beyond which cognitive functions reach satisfactory levels. This limit is the G3 level, i.e., 4899 steps a day. As in the study by Lee et al. [19], the number of 4400 steps per day is enough to prevent early death. Below these values, as demonstrated in the study, i.e., at the level of G1 or G2 activity (1245 and 2602 steps per day), there is no guarantee of the intended health impact of physical activity. So, it can be concluded that any amount of activity is important, but if it is not enough, it does not ensure that the health effects will be appropriate.

Referring to the above, in our study, statistically significant differences in cognitive functioning between groups with different physical activity relate primarily to the risk of depression, cognitive impairment, psychomotor speed and efficiency, auditory memory and learning efficiency, and dementia (Table 2). The advantage of this study is that it was already shown in a group of 110 women who practiced relatively simple physical activity. Compared to the study by Lee et al., it can be concluded that a larger number of participants will not diametrically affect the obtained results.

It should be emphasized that the age of the study participants was not a reason for decreased physical activity with age. On the other hand, the women differed between the groups in the value of the BMI index, the average value of which indicating that the entire research group was overweight [37]. However, when analyzing its values in individual groups (G1, G2, G3, and G4), it decreased with increasing physical activity. It is true that in none of the groups was the BMI value “normal”, but in the group with the highest PA (G4), the average value of the index was in the lower limit of the range, which proves that this group is striving for the “normal” value, while in the group with the highest PA (G4) and in the group with the lowest PA (G1), the mean BMI value indicates obesity of Iº.

In numerous studies, scientists have confirmed that a lack of physical activity or a small amount of itis conducive to obesity, i.e., a state in which the amount of energy supplied (in the form of food) significantly exceeds its consumption by the body and is characterized by an increase in the amount of adipose tissue. In contrast, obesity is a complex condition with serious social and psychological consequences. The results of our study confirm the influence of obesity on basic cognitive functions.

Moreover, the significant statistical differences between the groups with the lowest and the highest physical activity prove the influence of this activity on the BMI index, and thus on an overweight state and obesity. Many literature data indicate the positive effects of physical (exercise) activity on a correct bodyweight and BMI value [38,39,40], which was also confirmed in this study.

Diversified physical activity, both in terms of the steps taken during the day and the minutes spent, in individual groups indicates the awareness of the respondents on the importance and impact of physical activity on their health. These women fulfilled the WHO recommendations in the field of physical activity, devoting an average of 35.6 (group G1) to 178.9 (group G4) minutes for physical activity per day, which, by multiplying by each day of the week, gives an amount that meets the WHO recommendations. It can therefore be concluded that the research group consists of women who understand the importance of physical activity, which they declared in their activity diaries.

The obtained test results in the field of selected cognitive functions clearly show the positive effects of physical activity on the assessed parameters of depression, dementia, motor and psychomotor skills, and auditory memory. There were clearly significant differences between the groups of women, with less and more physical activity in all the parameters assessed.

The results obtained in the field of the absence of the symptoms of depression in over 90% of women in each group clearly show the impact of physical activity on the risk of depression, because in none of the groups were such symptoms found. Nevertheless, the group of women with the highest activity (G4) obtained a score half as high (2.5) as the group with the lowest activity (G1)—4.3.

In addition, in the group with the highest physical activity, no depression was found in any of the women. Such results have been confirmed in the literature on the subject, as well as in many epidemiological studies, which confirmed the influence of physical activity on the improvement of general well-being and mental health [41,42]. In studies by Dinas et al., the evidence suggested that physical exertion and physical activity have a beneficial effect on the symptoms of depression, comparable to the effects of antidepressants [43].

Interesting conclusions were drawn by Jerstad et al. from a study of 496 adolescent girls followed over a period of 6 years, in which they showed that physical activity significantly reduces the risk of future worsening of depression symptoms and the risk of severe and minor depression, but also interventions that increase physical activity can reduce the risk of depression in a high-risk population. Older people also constitute such a population [44].

Schuchs et al. [45] describe physical activity as an antidepressant drug. In meta-analysis of 49 prospective cohort studies encompassing 1,837,794 person-years found that people with high levels of physical activity had 17% lower odds of depression (OR = 0.83, CI = 0.79, 0.88) than people with low physical activity. Other meta-analyses have also found that low physical activity is associated with a greater risk of depression [46,47]. Other meta-analyses found that low cardiorespiratory fitness (CRF), an indicator of physical inactivity, was associated with a 64% higher risk of depression (HR = 1.64, CI = 1.29, 2.08) than high CRF across at least 3,540,450 person-years of data [48,49].

Similarly, the results of the MMSE test and the clock drawing test showed that physical activity influenced cognition. Women with more physical activity scored better in the conducted tests, while in the group with the highest activity, as many as 88% of the women did not have dementia, and only 12% had cognitive impairment without dementia. In this group of respondents, there was no case of mild dementia, and almost 100% of this group correctly performed the clock drawing test.

The group with the lowest physical activity (G1) had the highest percentage of diagnosed dementia, cognitive impairment, and mild dementia, as well as errors made in the clock test. More than half of this group had cognitive impairment and mild dementia. The above conclusions have been confirmed by numerous publications around the world.

Interesting research was carried out by de Bruijn et al. [50], who examined the relationship between physical activity and the risk of dementia over a 14-year follow-up period in 4406 elderly people (age range 61–97) from the prospective population-based Rotterdam Study and found that higher levels of physical activity are associated with a lower risk of dementia.

A meta-analysis of studies on the influence of aerobic exercise on the improvement of cognitive functioning in elderly people diagnosed with fair cognitive disorders also showed that this type of training significantly influenced the improvement of, among other things, MMSE tests [51]. The positive effects of aerobic training (consisting of regular walks) on cognitive functions were confirmed by studies conducted among women aged 70–80 suffering from mild cognitive impairment [52].

A meta-analysis of longitudinal studies on the long-term effects of regular physical activity on the cognitive processes of people over 65 years of age showed that regular moderate exercise was associated with a reduction in cognitive impairment by as much as 50% and reduced the risk of dementia or delayed the onset of dementia symptoms [53].

Our study also showed a positive effect of physical activity on auditory memory, as the results obtained clearly indicate that women with higher physical activity learned more words and remembered more words after a delay of 30 min, and women with the lowest levels of activity had poor memory because they could remember the fewest words.

Our conclusions also concur with earlier studies, such as those of Lambourne and Tomporowski [54] or McMorris et al. [55], who proved that moderate one-time physical activity improves the ability to remember and shortens the time of making decisions. Longitudinal studies conducted in the United Kingdom in a group of 10,652 elderly adults showed that people who were not physically active obtained worse results in memory tests than active people [56].

Numerous studies conducted with the participation of healthy people and those showing signs of cognitive deterioration or even advanced dementia have proved that physical activity maintains greater cognitive performance, while in the field of mental health, physical effort reduces stress and anxiety, improves mood, and thus has an antidepressant effect.

Our research, however, shows that there is a clear amount of physical activity that guarantees an adequate level of cognitive maintenance, about 5000 steps a day of appropriate intensity, which can guarantee that cognitive functioning will be at the appropriate level. In addition, such activity is practical and convenient for older people, who can themselves improve their awareness that the right activity affects depression or dementia that may come later in life.

However, it should be borne in mind that even regular physical activity is not able to replace traditional pharmacological treatment or psychotherapy but is a preventive and supportive factor in the treatment of many disorders related to the functioning of the brain.

## 5. Conclusions

The research results presented in this paper show a strong correlation between undertaking physical exercise and better mental functioning, both in cognitive and emotional terms, in women over 60 years of age, while approximately 5000 steps a day at the right intensity are required to be more mentally fit. These results show that about 5000 steps a day is enough to see a positive effect on the mental health and cognitive functioning of this group in the elderly population. Thus, you do not need a lot of activity (10 thousand steps) as G3 (4899 steps a day) is enough to ensure proper cognitive processes.Regular physical activity can be an important and powerful protective factor for cognitive impairment and dementia in the elderly, but too little activity (less than 2000 steps a day) does not guarantee a clearly positive effect on health. Thus, physical activity of an appropriate amount and intensity should be promoted not only for the prevention of cardiovascular or chronic diseases and many other diseases of civilization, but also because of its impact on mental health.Low physical activity contributes to obesity, which is associated with cognitive impairment in the elderly.Programs aimed at increasing physical fitness can serve as a tool to improve cognitive functions, including memory. Additionally, increased physical activity can be used as an adjunct to the treatment of depression.The presented research results show the health-promoting effects of physical activity in the right amount, and may constitute the basis for more in-depth and wider research (e.g., on a group of children and adolescents) in order to introduce this phenomenon and understand its mechanism, and in the longer term to develop optimal prophylactic and therapeutic strategies; thus, exercise should be considered a basic, relatively cheap drug of the 21st century.

## Figures and Tables

**Table 1 ijerph-18-07549-t001:** Characteristics of the research group and physical activity in the classified groups.

Characteristic	Total	Average of Mean Steps per Day as Mean ± SD (Range), Median			
		Group 1 (G1) (Lowest)	Group 2 (G2)	Group 3 (G3)	Group 4 (G4) (Highest)
Total Participants, No	110 (100%)	23 (20.9%)	32 (29.1%)	38 (34.6%)	17 (15.4%)
Age, year	67.3 ± 6.1	68.9 ± 6.9	68.0 ± 5.9	66.5 ± 6.2	65.6 ± 3.6
	(60.0–91.0)	(60.0–91.0)	(60.0–84.0)	(60.0–91.0)	(61.0–72.0)
	67.0	67.0	68.0	65.0	66.0
Statistically significant differences between means		Not significant			
BMI	28.1 ± 4.7	30.3 ± 4.9	28.7 ± 5.0	27.0 ± 4.4	26.3 ± 2.9
	(18.4–42.5)	(19.5–38.3)	(21.4–42.5)	(18.4–36.0)	(19.9–32.7)
	27.3	31.1	28.1	27.1	25.9
Statistically significant differences between means (*p*)		G1–G3 (0.0140) G1–G4 (0.0033) G2–G4 (0.0404)			
Smokers/former smokers, No (%)	18/23 (16/21)	1/5 (4/22)	9/2 (28/6)	5/11 (13/29)	3/5 (18/29)
Statistic χ^2^		39.57			
*p*		0.0000			
Higher education, No (%)	41 (37)	9 (39)	10 (31)	16 (42)	6 (35)
Statistic χ^2^		2.96			
*p*		0.3982			
Steps/day	4462 ± 3465	1245 ± 336	2602 ± 336	4899 ± 1197	11,335 ± 2434
	(746–16,750)	(746–1788)	(2080–3120)	(3200–7450)	(8483–16,750)
	3160.0	1193	2558	4542	10,087
Minutes/day	92.8 ± 52.8	35.6 ± 10.7	68.5 ± 17.4	109.4 ± 33.2	178.9 ± 35.9
	(14.3–240.0)	(14.3–55.0)	(35.7–102.9)	(64.3–188.6)	(121.4–240.0)
	81.1	30.0	68.9	98.6	172.9
Intensity of physical activity (steps/minutes)	45.3 ± 13.8	36.1 ± 8.0	40.7 ± 8.0	46.0 ± 11.4	59.3 ± 19.2
	(24.4–91.6)	(26.8–61.5)	(30.7–61.6)	(24.4–61.5)	(13.8–91.6)
	40.4	33.6	38.1	50.8	60.3
Statistically significant differences between means (*p*)		G1–G3 (0.0001) G1–G4 (0.0000) G2–G3 (0.0044) G2–G4 (0.0000) G3–G4 (0.0000)			

**Table 2 ijerph-18-07549-t002:** Characteristics of the cognitive function tests.

Characteristic	Total	Average of Mean Steps per Day as Mean ± SD (Range), Median			
		Group 1 (G1) (Lowest)	Group 2 (G2)	Group 3 (G3)	Group 4 (G4) (Highest)
GDS	3.9 ± 3.3	4.3 ± 3.3	4.2 ± 3.5	3.9 ± 3.4	2.5 ± 2.5
	(0.0–14.0)	(0.0–12.0)	(0.0–12.0)	(0.0–14.0)	(0.0–8.0)
	3.0	3.0	4.0	3.0	2.0
Statistically significant differences between means (*p*)		G1–G4 (0.0430)			
		G2–G4 (0.0472)			
MMSE	27.3 ± 2.4	26.1 ± 2.5	27.3 ± 2.4	27.6 ± 2.3	28.5 ± 1.6
	(21.0–30.0)	(21.0–30.0)	(22.0–30.0)	(22.0–30.0)	(24.0–30.0)
	28.0	26.0	28.0	28.0	29.0
Statistically significant differences between means (*p*)		G1–G3 (0.0280) G1–G4 (0.0012)			
Motor speed	20.6 ± 4.3	19.7 ± 5.0	19.4 ± 3.2	21.3 ± 4.0	22.3 ± 4.1
	(10.0–31.0)	(12.0–26.0)	(14.0–26.0)	(10.0–29.0)	(11.0–30.0)
	20.0	18.0	19.0	21.0	22.0
Statistically significant differences between means (*p*)		G1–G3 (0.0473) G1–G4 (0.0341)			
		G2–G3 (0.0416) G2–G4 (0.0281)			
Psychomotor speed	5.7 ± 1.8	5.0 ± 1.9	5.4 ± 1.6	5.9 ± 1.5	7.0 ± 1.6
	(2.0–10.0)	(2.0–9.0)	(3.0–10.0)	(2.0–9.0)	(4.0–10.0)
	6.0	5.0	5.0	6.0	7.0
Statistically significant differences between means (*p*)		G1–G3 (0.0281) G1–G4 (0.0006)			
		G2–G4 (0.0036)			
		G3–G4 (0.0361)			
Luria’s auditory memory test	36.1 ± 6.2	32.5 ± 6.1	36.5 ± 5.1	37.5 ± 6.8	37.1 ± 4.6
	(12.0–48.0)	(18.0–43.0)	(27.0–47.0)	(12.0–48.0)	(25.0–43.0)
	37.0	32.0	36.0	38.0	37.0
Statistically significant differences between means (*p*)		G1–G2 (0.0152) G1–G3 (0.0051) G1–G4 (0.0124)			
Recall test	5.8 ± 2.4	4.3 ± 2.0	5.4 ± 2.4	6.9 ± 2.2	6.8 ± 1.7
	(0.0–10.0)	(0.0–7.0)	(0.0–10.0)	(0.0–10.0)	(3.0–9.0)
	6.0	5.0	5.0	7.0	7.0
Statistically significant differences between means (*p*)		G1–G3 (0.0000) G1–G4 (0.0002) G2–G3 (0.0131) G2–G4 (0.0020)			
Clock test	2.7 ± 0.7	2.3 ± 0.9	2.7 ± 0.7	2.9 ± 0.4	2.8 ± 0.7
	(0.0–3.0)	(0.0–3.0)	(0.0–3.0)	(1.0–3.0)	(0.0–3.0)
	3.0	3.0	3.0	3.0	3.0
Statistically significant differences between means (*p*)		G1–G3 (0.0064) G1–G4 (0.0459)			

**Table 3 ijerph-18-07549-t003:** The percentage distribution and statistics of the results of the depression and dementia tests performed among the studied women.

TEST	Description	Average of Mean Steps per Day as Mean ± Sd (Range), Median			
		Group 1 (G1) (Lowest)	Group 2 (G2)	Group 3 (G3)	Group 4 (G4) (Highest)
GDS	without depression	91	94	92	100
	mild depression	9	6	8	0
Statistic χ^2^		6.04			
*p*		0.1095			
MMSE	no dementia	48	66	61	88
	cognitive impairment without dementia	30	25	34	12
	mild dementia	22	9	5	0
Statistic χ^2^		49.82			
*p*		0.0000			
Clock test	correct	57	75	95	94
	with errors	43	25	5	6
Statistic χ^2^		61.50			
*p*		0.0000			

**Table 4 ijerph-18-07549-t004:** The correlations between physical activity, expressed in the number of daily steps taken and the intensity of activity, and BMI and other assessed parameters of the health status of seniors.

Steps/Day	BMI	r = −0.2561	Intensity of Physical Activity (Steps/Minutes)	BMI	r = −0.2337
		*p* = 0.0069			*p* = 0.0139
	GDS	r = −0.1653		GDS	r = −0.2806
		*p* = 0.8439			*p* = 0.0030
	MMSE	r = 0.2674		MMSE	r = 0.1681
		*p* = 0.0047			*p* = 0.0792
	Motor speed	r = 0.2624		Motor speed	r = 0.2030
		*p* = 0.0056			*p* = 0.0334
	Psychomotor speed	r = 0.3347		Psychomotor speed	r = 0.2867
		*p* = 0.0003			*p* = 0.0024
	Luria’s auditory memory test	r = 0.1913		Luria’s auditory memory test	r = 0.0902
		*p* = 0.0453			*p* = 0.3487
	Recall test	r = 0.2260		Recall test	r = 0.1293
		*p* = 0.0176			*p* = 0.1782
	Clock test	r = 0.2106		Clock test	r = 0.1838
		*p* = 0.0272			*p* = 0.0546

## Data Availability

The datasets generated during and/or analyzed during the currentstudy are available from the corresponding author on reasonable request.

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
