# Peer review of "The Association between Physical Activity and Selected Parameters of Psychological Status and Dementia in Older Women"

_ijerph, 2021, doi:10.3390/ijerph18147549_

Round 1
Reviewer 1 Report
Your study looks for differences in the relationship between physical activity and physical and psychological parameters in women more than 60 years old that are involved in regular physical activity.
From what I understand, it's an observational study with no intervention
But how were the participants recruited? What were the admission/exclusion criteria used? The fact that the participants had no health problems was intentional? Why was the study done on a uni gender population?
For how long was the study made in each participant, there's some information on "assessment of physical activity," "record their physical activities each day of the week," "repeated three times," has there been an interval between the repetitions?
In the conclusions, eventually changing the"G3" with the number of steps. gives a clearer message
Author Response
Thank you very much for any comments and suggestions to improve my manuscript. In my manuscript, I put corrections in line with the Reviewers' guidelines. I put the changes made in red. I added information on how to recruit study participants, admission/exclusion criteria, gender selection and how to measure physical activity. In the conclusions, I added „G3” with the number of steps, the advantages of this study and influence the number of participants.

Reviewer 2 Report
The paper presents a study that shows the impact of physical activity on selected cognitive functions in physically active women over 60 years of age.
In order to achieve the study, a group of 110 women over 60 years of age was considered, who were divided into 4 groups based on the intensity of their physical activity.
The women included in this analysis had an average age of 67.3 and were divided into 4 groups (G1 - lowest activity, G2, G3, G4 - highest activity) that were statistically different in terms of physical activity.
Tests show that about 5,000 steps a day are enough to see a positive effect on mental health and cognitive functioning in women over 60 to ensure cognitive processes.
I would recommend highlighting the advantages of this study compared with other similar research in the field. Also, if the number of participants can influence the results obtained, it would be important to know.
Author Response
Thank you very much for any comments and suggestions to improve my manuscript. In my manuscript, I put corrections in line with the Reviewers' guidelines. I put the changes made in red. I added information on how to recruit study participants, admission/exclusion criteria, gender selection and how to measure physical activity. In the conclusions, I added „G3” with the number of steps, the advantages of this study and influence the number of participants.

This manuscript is a resubmission of an earlier submission. The following is a list of the peer review reports and author responses from that submission.